# Targeted surveillance strategies for efficient detection of novel antibiotic resistance variants

**Allison L Hicks[1], Stephen M Kissler[1], Tatum D Mortimer[1], Kevin C Ma[1], George Taiaroa[2], Melinda Ashcroft[2], Deborah A Williamson[2], Marc Lipsitch[1,3], Yonatan H Grad[1,4]***

[1]Department of Immunology and Infectious Diseases, Harvard T.H. Chan School of Public Health, Boston, United States; [2]Department of Microbiology and Immunology, The University of Melbourne at The Peter Doherty Institute for Infection and Immunity, Melbourne, Australia; [3]Center for Communicable Disease Dynamics, Department of Epidemiology, Harvard T.H. Chan School of Public Health, Boston, United States; [4]Division of Infectious Diseases, Department of Medicine, Brigham and Women's Hospital, Harvard Medical School, Boston, United States

**Abstract** Genotype-based diagnostics for antibiotic resistance represent a promising alternative to empiric therapy, reducing inappropriate antibiotic use. However, because such assays infer resistance based on known genetic markers, their utility will wane with the emergence of novel resistance. Maintenance of these diagnostics will therefore require surveillance to ensure early detection of novel resistance variants, but efficient strategies to do so remain undefined. We evaluate the efficiency of targeted sampling approaches informed by patient and pathogen characteristics in detecting antibiotic resistance and diagnostic escape variants in *Neisseria gonorrhoeae*, a pathogen associated with a high burden of disease and antibiotic resistance and the development of genotype-based diagnostics. We show that patient characteristic-informed sampling is not a reliable strategy for efficient variant detection. In contrast, sampling informed by pathogen characteristics, such as genomic diversity and genomic background, is significantly more efficient than random sampling in identifying genetic variants associated with resistance and diagnostic escape.

**\*For correspondence:**
ygrad@hsph.harvard.edu

## Introduction

Nucleic acid-based diagnostics that enable rapid pathogen identification and prediction of drug susceptibility profiles can improve clinical decision-making, reduce inappropriate antibiotic use, and help address the challenge of antibiotic resistance (*McAdams et al., 2019*; *Fingerhuth et al., 2017*; *Tuite et al., 2017*). However, the sensitivity of such diagnostics may be undermined by undetected genetic variants (*André et al., 2017*; *Berhane et al., 2018*; *Herrmann et al., 2008*; *Guglielmino et al., 2019*; *Whiley et al., 2011*; *Golparian et al., 2012*; *Bruisten et al., 2004*; *Lee et al., 2018a*; *Marks et al., 2018*). Pathogen surveillance programs aimed at early detection of novel variants are crucial to ensuring the clinical utility and sustainability of these diagnostics.

Use of traditional nucleic acid amplification tests (NAATs) for pathogen identification and genotype-based diagnostics for antibiotic resistance can select for genetic variants that escape detection (*Smid et al., 2019*). Mutations and/or deletions at the NAAT target locus that cause an amplification failure have arisen in *Neisseria gonorrhoeae*, *Chlamydia trachomatis*, *Staphylococcus aureus*, and *Plasmodium falciparum*, resulting in false negative diagnostic errors only detected when using

another diagnostic platform (*Berhane et al., 2018*; *Herrmann et al., 2008*; *Guglielmino et al., 2019*; *Lee et al., 2018a*). Diagnostic escape associated with genotype-based diagnostics for antibiotic resistance are the result of resistance-conferring variants (*e.g.*, mutations or accessory genes) not accounted for in the diagnostic's panel of resistance markers (*André et al., 2017*) and require phenotypic testing to be uncovered.

We recently presented a framework to quantify the sampling rate for early detection of novel antibiotic resistance variants, defining the number of isolates that would need to undergo confirmatory phenotyping from those predicted by genotype to be susceptible (*Hicks et al., 2019*). Underlying this model are assumptions of unbiased sampling across a population and independence among all isolates. However, these assumptions may not hold in practice, as some subsets of the population (*e.g.*, demographics and/or geographic regions) may be more likely to be sampled than others, and clonal transmission may result in repeated sampling of closely related isolates (*Rempel and Laupland, 2009*; *Unemo et al., 2019*; *Hutinel et al., 2019*; *Van Goethem et al., 2019*). The real-world application of this model may also be challenging for pathogens with high case incidence, such as *N. gonorrhoeae*, as the cost of phenotyping required by this model for timely detection of novel resistance variants is likely to be high (*Hicks et al., 2019*).

Implementing a practical surveillance system thus requires improving efficiency over unbiased testing by prioritizing samples in which novel diagnostic escape variants are most likely to be found. There are numerous hypotheses for how to focus sampling and most quickly identify these variants. Novel variants may be more likely to emerge or spread in certain anatomical niches, demographics, or geographic regions (*Lewis, 2013*; *Collignon et al., 2018*; *Frost et al., 2019*; *Hernando Rovirola et al., 2020*), some of which may be systematically under-sampled (*Kirkcaldy et al., 2019*) and thus may provide a basis for sampling priority. Data on such characteristics may be obtained from metadata recorded during clinical encounters. Alternatively, they may be inferred from pathogen genomic data. Isolates or clades that are genetically divergent from the majority of isolates in a population may reflect travelers, their contacts, or otherwise under-sampled lineages (*Perrin et al., 2003*; *Pham Thanh et al., 2016*; *Kingsley et al., 2009*; *Mac Aogáin et al., 2016*). Some pathogen genomic backgrounds may be more conducive to the evolution of novel resistance mechanisms (*Borrell and Gagneux, 2011*), and markers of these genomic backgrounds (*e.g.*, variants associated with a range of resistance mechanisms and/or resistance to other drugs) may help improve sampling efficiency. Similarly, given historical patterns of antibiotic use, novel resistance may emerge on a background of existing resistance (*Gould and MacKenzie, 2002*). Thus, genetic markers of resistance to certain drugs may facilitate identification of lineages more likely to have experienced selective pressures leading to emergence of novel resistance variants.

Here, we test the performance of sampling strategies informed by these hypotheses using *N. gonorrhoeae* surveillance data. *N. gonorrhoeae* offers a useful model, given the increasing drug resistance and recent focus on developing sequence-based resistance diagnostics (*Fingerhuth et al., 2017*; *Hook and Kirkcaldy, 2018*). We present targeted sampling approaches informed by patient (i.e., demographics, anatomical site of isolate collection, geographical region, recent travel history, or sex worker status) and pathogen (i.e., phylogenetic or genomic background) information. We assess the efficiency of each of these strategies to detect rare (<10% prevalence) resistance variants associated with current or recent first-line recommended antibiotics (i.e., azithromycin [AZM] and extended spectrum cephalosporins [ESCs]), as well as rare genetic variants associated with diagnostic escape, across five genomic surveys with various demographic, geographic, and temporal ranges. We show that phylogeny- and genomic background-aware sampling approaches can increase the detection efficiency of known variants over random sampling, whereas patient feature-based sampling approaches do not. Our results suggest that implementation of such targeted sampling approaches into surveillance programs may reduce the number of cases of novel resistance that occur before it is detected, as well as the resources required to undertake surveillance, compared to random sampling of a population.

# Results

## Composition of the datasets

The datasets (*Table 1*) were biased across patient demographics and/or geographic regions (*Supplementary files 1* and *2*A). Isolates from men and men who have sex with men (MSM) were overrepresented in datasets 1 and 2 compared to overall gonorrhea incidence in men and MSM in the US and Australia, respectively, during the study periods (*Supplementary file 2A*, p<0.001 for both datasets by chi-squared test of men vs. women and MSM vs. non-MSM in dataset vs. reported incidence). Dataset 4 was comprised exclusively of isolates from men (*Yahara et al., 2018*). While it is difficult to estimate the prevalence of pharyngeal gonococcal infections, as they tend to be asymptomatic (*Wiesner et al., 1973*), pharyngeal isolates represented 4% and 18% of isolates with reported anatomical site of collection in datasets 1 and 2, respectively. This suggests either sampling bias across anatomical sites in at least one of the datasets or substantial variation across the two study populations in prevalence of pharyngeal gonococcal infections. Similarly, the geographic distribution of isolates in dataset 3 was significantly different from the reported case incidence across countries (*Supplementary file 2A*, p<0.001 by chi-squared test of prevalence for each of the countries in dataset 3 vs. the reported overall incidence for each of the countries).

## Targeted sampling based on patient characteristics

We investigated whether sampling evenly across demographic groups (demography-aware sampling), anatomical sites of isolate collection (niche-aware sampling), and geographic regions (geography-aware sampling) increased detection efficiency of resistance variants by ameliorating some of the demographic, niche, or geographic sampling biases. We further investigated whether preferentially sampling patients with recent overseas sexual encounters or recent sex work, two characteristics hypothesized to be associated with the introduction and/or increased transmission of resistance (*Lewis, 2013*; *Frost et al., 2019*; *Hernando Rovirola et al., 2020*), increased the detection efficiency of resistance variants. To do so, we simulated and compared the detection efficiency of three genetic resistance variants (*Table 2*) using each of these targeted sampling strategies and random sampling.

The detection efficiency was not improved by demography-, niche-, geography-aware sampling compared to random sampling for any of the resistance variants (*Supplementary file 2B*, *Figure 1*). In several cases, detection efficiency significantly decreased in demography- or geography-aware sampling compared to random sampling, reflecting enrichment of the resistance variant in the overrepresented demographic or geographic region. However, no significant association between a given resistance variant and demographic group was observed across both dataset 1 and dataset 2, and no demographics or geographic regions were significantly enriched for all variants (*Supplementary file 1*), suggesting that preferential sampling of any of these demographics or geographic regions would not be a reliable strategy for increasing novel variant detection efficiency. For example, while *penA* XXXIV was significantly enriched in MSM compared to men who have sex with women and women who have sex with men (MSW/WSM) in dataset 2 (p<0.003, Fisher's exact test), there was no significant difference in the proportions of MSM and MSW/WSM with *penA* XXXIV in

**Table 1.** Summary of datasets.

| Dataset | Temporal range | N$_{isolates}$ | Geographic range | Metadata available | SRA study ID/Reference |
|---|---|---|---|---|---|
| 1 | 2011–2015 | 896 | New York, NY, US | Gender, sexual behavior, anatomical site of isolation | ERP011192 (*Mortimer et al., 2020*) |
| 2 | 2016–2017 | 2186 | Victoria, Australia | Gender, sexual behavior, anatomical site of isolation, travel history, sex worker status | SRP185594 (*Williamson et al., 2019*) |
| 3 | 2013 | 1054 | Europe | Country of sample collection | ERP010312 (*Harris et al., 2018*) |
| 4 | 2015 | 244 | Japan | Prefecture of sample collection | DRP004052 (*Yahara et al., 2018*) |
| 5 | 2014–2015 | 398 | New Zealand | N/A | SRP111927 (*Lee et al., 2018b*) |

**Table 2.** Summary by dataset of the prevalence and distribution of the genetic markers of resistance and resistance phenotypes tested.

| Variant | | Genetic | | | Phenotypic | |
| --- | --- | --- | --- | --- | --- | --- |
| | | **RplD G70D** | **23S rRNA C2611T (2–4 alleles)** | **penA XXXIV** | **CRO-RS (≥0.12 µg/mL)** | **CFX-R (>0.25 µg/mL)** |
| Drug | | AZM (*Grad et al., 2016*) | AZM (*Lk et al., 2002*) | ESCs (*Grad et al., 2014*) | N/A | N/A |
| Prevalence of variant in dataset | 1 | 10.04%* | 0.11% | 5.25% | 1.47% | 0.11% |
| | 2 | 1.14% | 1.24% | 1.69% | 0% | 0% |
| | 3 | 2.47% | 0.95% | 15.68%* | 1.04% | 0.76% |
| | 4 | 11.07%* | 1.23% | 0.41% | 6.56% | 8.20% |
| | 5 | 0.75% | 0.50% | 2.26% | 0.25% | 0% |
| Phylogenetic D statistic for variant in dataset | 1 | −0.18 | 17.50 | −0.29 | N/A | N/A |
| | 2 | −0.10 | 0.46 | −0.24 | N/A | N/A |
| | 3 | 0.05 | 0.30 | −0.20 | N/A | N/A |
| | 4 | −0.16 | 1.83 | 1.81 | N/A | N/A |
| | 5 | 0.83 | 1.12 | −0.15 | N/A | N/A |

*Given the >10% prevalence of RplD G70D in datasets 1 and 4 and *penA* XXXIV in dataset 3, these variants were excluded from sampling simulations. AZM, azithromycin; ESC, extended-spectrum cephalosporin; CRO-RS, ceftriaxone reduced susceptibility; CFX-R, cefixime resistance.

dataset 1 (p=0.461, Fisher's exact test). Similarly, while the AZM-R-associated RplD G70D mutation in dataset 3 was at highest prevalence in patients from Malta and Greece (10% and 6.25%, respectively) and absent from patients from Denmark, the AZM-R-associated 23S C2611T variant was at highest prevalence in patients from Denmark (5.45%) and absent from patients from Malta or Greece.

Isolates from patients with recent overseas sex were associated with significantly longer terminal branches compared to patients that had only engaged in sex locally (*Figure 1—figure supplement 1*), in support of the hypothesis that international travel may be associated with the importation of novel or divergent strains, or, more generally, that isolates from travelers may be more likely to be associated with under-sampled lineages. Preferentially sampling from patients with recent overseas sex significantly improved detection efficiency of the RplD G70D mutation and the *penA* XXXIV allele, as these were at marginally higher prevalence in isolates from patients with recent overseas sex compared to those from patients who had only engaged in sex locally (3.03% overseas vs. 0.98% local and 2.02% overseas vs. 1.67% local, respectively, p=0.090 and 0.683, respectively, by Fisher's exact test for both variants). In contrast, the 23S C2611T mutation was exclusively present in isolates from patients who had engaged in sex locally (*Supplementary files 1* and *2*C). Similarly, while the 23S C2611T mutation was marginally enriched in isolates from patients who had engaged in recent sex work compared to patients who had not (2.33% in sex workers vs. 1.31% in non-sex workers, p=0.327 by Fisher's exact test), and thus preferentially sampling from sex workers significantly improved detection efficiency of this variant compared to sampling from the full patient population, detection efficiencies for the RplD G70D mutation and the *penA* XXXIV allele were not significantly improved by preferentially sampling from sex workers (*Supplementary files 1* and *2* C).

Together, these results suggest that while targeted sampling based on patient characteristics may increase detection efficiency of some novel variants, it is difficult to predict which groups to target for all potential novel variants.

## Targeted sampling based on genetic diversity

To assess whether preferential sampling of lineages that are divergent from those that have been previously sampled may increase detection efficiency of genetic resistance variants over random sampling, we simulated phylogeny-aware sampling using two methods: 1) maximization of the phylogenetic distance covered with each isolate sampled (distance maximization) and 2) even sampling across phylogenetic lineages (clonal group).

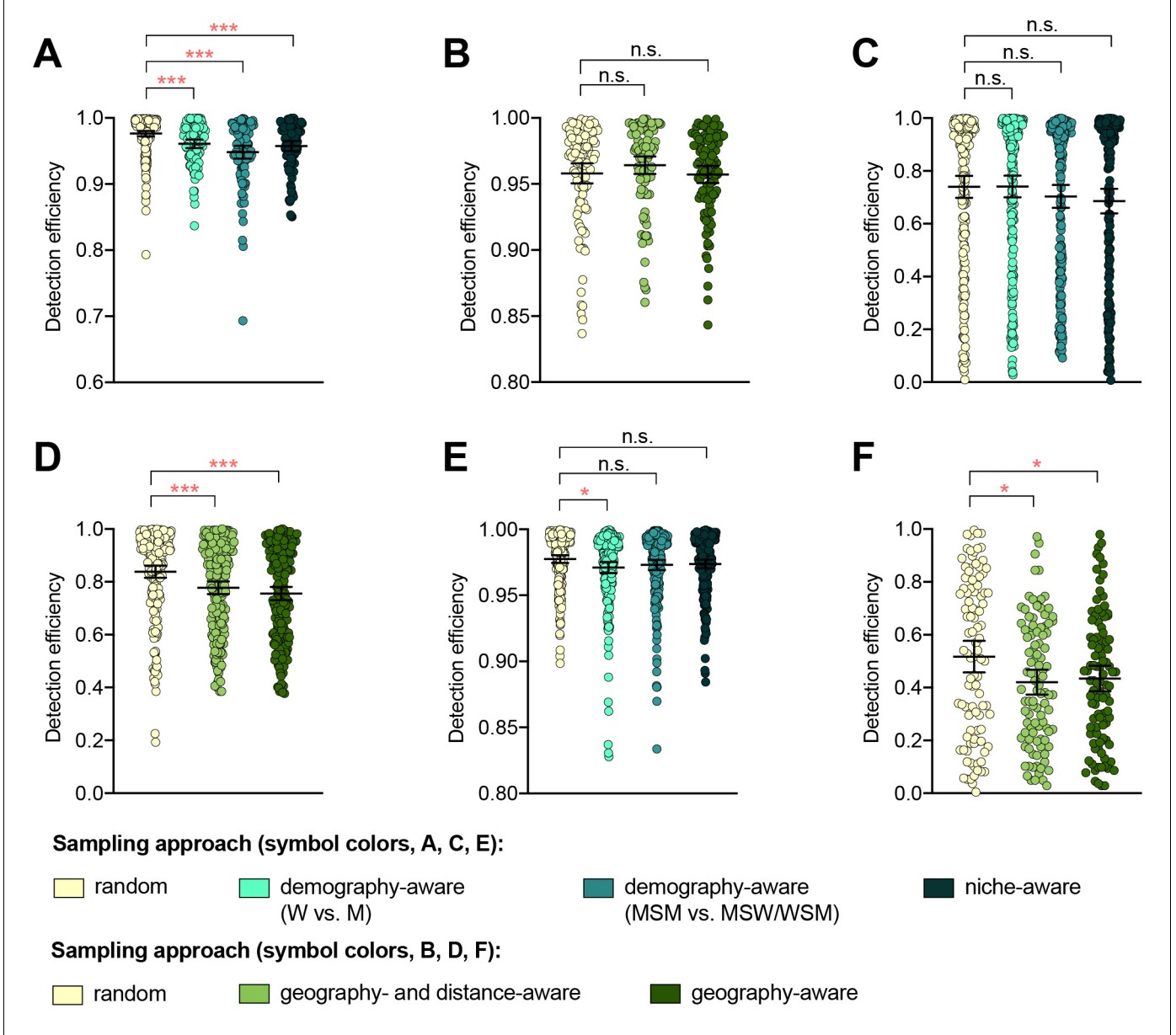

**Figure 1.** The impact of demography-, niche-, and geography-aware sampling on the detection efficiency of genetic resistance variants. Dot plots showing the detection efficiency (with lines indicating the mean and 95% confidence intervals from 100 simulations) for resistance variants RplD G70D (A–B), 23S rRNA C2611T (C–D), and *penA* XXXIV (E–F) in datasets 1 and 2. In datasets 1 and 2, targeted sampling was informed by demographic (gender and sexual behavior) and anatomical site of isolate collection (niche) information (A, C, and E), and in datasets 3 and 4, targeted sampling was informed by country or prefecture of sample collection (B, D, and F). Dot colors indicate the sampling approach, and asterisks indicate a significant difference (p<0.05 by Mann-Whitney U test) in detection efficiency between the demography-, niche- or geography-aware approach compared to random sampling (*p<0.05, **p<0.01, ***p<0.001; red asterisks indicate significantly lower detection efficiency of demography- or geography-aware approaches compared to random sampling). Note that sampling simulations were not performed for RplD G70D in datasets 1 and 4 or for *penA* XXXIV in dataset 3 as prevalence of the variants in these datasets was >10%. n.s., not significant at α = 0.05; M, men; W, women; MSM, men who have sex with men; MSW, men who have sex with women; WSM, women who have sex with men.

The online version of this article includes the following figure supplement(s) for figure 1:

**Figure supplement 1.** Isolates from patients with travel-associated gonorrhea are associated with longer terminal branches compared to patients with locally-acquired gonorrhea.

While the distance maximization approach increased detection efficiency compared to random sampling for some variants, there were numerous instances in which this approach, which led to preferential sampling of isolates associated with long branches, substantially decreased detection efficiency (*Figure 2*, *Supplementary file 2D*).

The clonal group sampling approach prevents repeated sampling of very closely related isolates until all unique phylogenetic clusters have been sampled. Thus, for both rare variants that are clonally distributed and rare variants that are more randomly dispersed throughout the phylogeny (*e.g.*, *penA* XXXIV and 23S rRNA C2611T mutations, respectively, *Table 2*), this approach increases detection efficiency in cases where 1) there is substantial clonality among isolates and 2) a substantial proportion of variant-positive isolates do not occur in clonal lineages dominated by variant-negative isolates (*Figure 2E*). In such datasets, effectively collapsing large variant-negative lineages into a single representative increases the effective prevalence of the variants and thus the detection efficiency of the clonal group approach compared to random sampling. The clonal group sampling approach significantly decreased detection efficiency in only one instance (i.e., the 23S rRNA C2611T variant in dataset 4, *Supplementary file 2D*), where all isolates with the variant appeared in large clonal lineages of predominately variant-negative isolates (*Figure 2D*).

In cases where the clonal group sampling approach did not perform better than random sampling, adjusting the threshold for clonal grouping and/or a marginal increase in the prevalence of variant-positive isolates could elevate the relative performance of this targeted approach. We chose 134 SNPs as an example threshold for clonal grouping, as it represents the lower 95% confidence interval of the mean of SNP distances between each CFX-R resistant and the closest susceptible isolate in datasets 1–5 (see Methods). In the case of the 23S rRNA C2611T variant in dataset 4, the average prevalence of the variant across clonal groups (i.e., the total number of variant-positive isolates, counting each variant-positive isolate as [1 / [1 + the total number of additional isolates that are ≤134 SNPs of the isolate]], divided by the number of clonal groups) is 0.005, lower than the actual prevalence of 0.012. However, if the threshold for clonal grouping was lower in this instance (*e.g.*, 50 SNPs), the effective prevalence of the variants would be 0.020, greater than the actual prevalence of 0.012. Similarly, using the 134 SNP threshold, if one additional isolate that was >134 SNPs from any other isolates in this dataset had the 23S rRNA C2611T mutation, the average prevalence of the variant across clonal groups would be 0.036, greater than the actual prevalence of 0.016, and thus the clonal group approach would outperform random sampling.

To further assess the performance of phylogeny-aware sampling in the context of rare genetic variants that may have emerged in response to diagnostic pressure, we simulated random and phylogeny-aware sampling to assess detection efficiency of an additional set of variants. Specifically, we assessed a panel of *N. gonorrhoeae* diagnostic escape variants: the 16S rRNA C1209A mutation, the *N. meningitidis*-like *porA*, and the *cppB* deletion, all of which have been previously associated with diagnostic failure (*Guglielmino et al., 2019*; *Whiley et al., 2011*; *Golparian et al., 2012*; *Bruisten et al., 2004*) and were present in one or more of datasets 1–5 at low prevalence (*Table 3*). The G168A mutation in the primer binding region of DR-9A, the target of the COBAS 4800 CT/NG (Roche) diagnostic, has not previously been documented but was present in 0.1% of strains from dataset 2. All of the diagnostic-associated variants assessed appeared in divergent backgrounds and were thus detected more efficiently by phylogeny-aware sampling compared to random sampling (*Figure 2F–I*, *Supplementary file 2E*). Like the results from the simulations based on resistance variants, the distance maximization approach maximized detection efficiency for some of the diagnostic-associated variants, but superiority of this approach to random sampling was not consistent across all variants. However, the clonal group approach performed significantly better than random sampling for all diagnostic-associated variants across all datasets.

The relative performance of the clonal group sampling approach compared to random sampling was generally consistent across multiple thresholds based on pseudogenomes (i.e.,≤134 SNPs, ≤422 SNPs, and fastBAPS groups); relative performance of clonal group sampling using MLSTs, however, was less consistent and was significantly worse than random sampling for several variants (*Figure 2—figure supplement 1*, *Supplementary file 2*D-E). Together, these results suggest that preferentially sampling isolates that, based on whole genome sequencing (WGS), are phylogenetically divergent from those that have previously been sampled may increase detection efficiency of novel resistance variants.

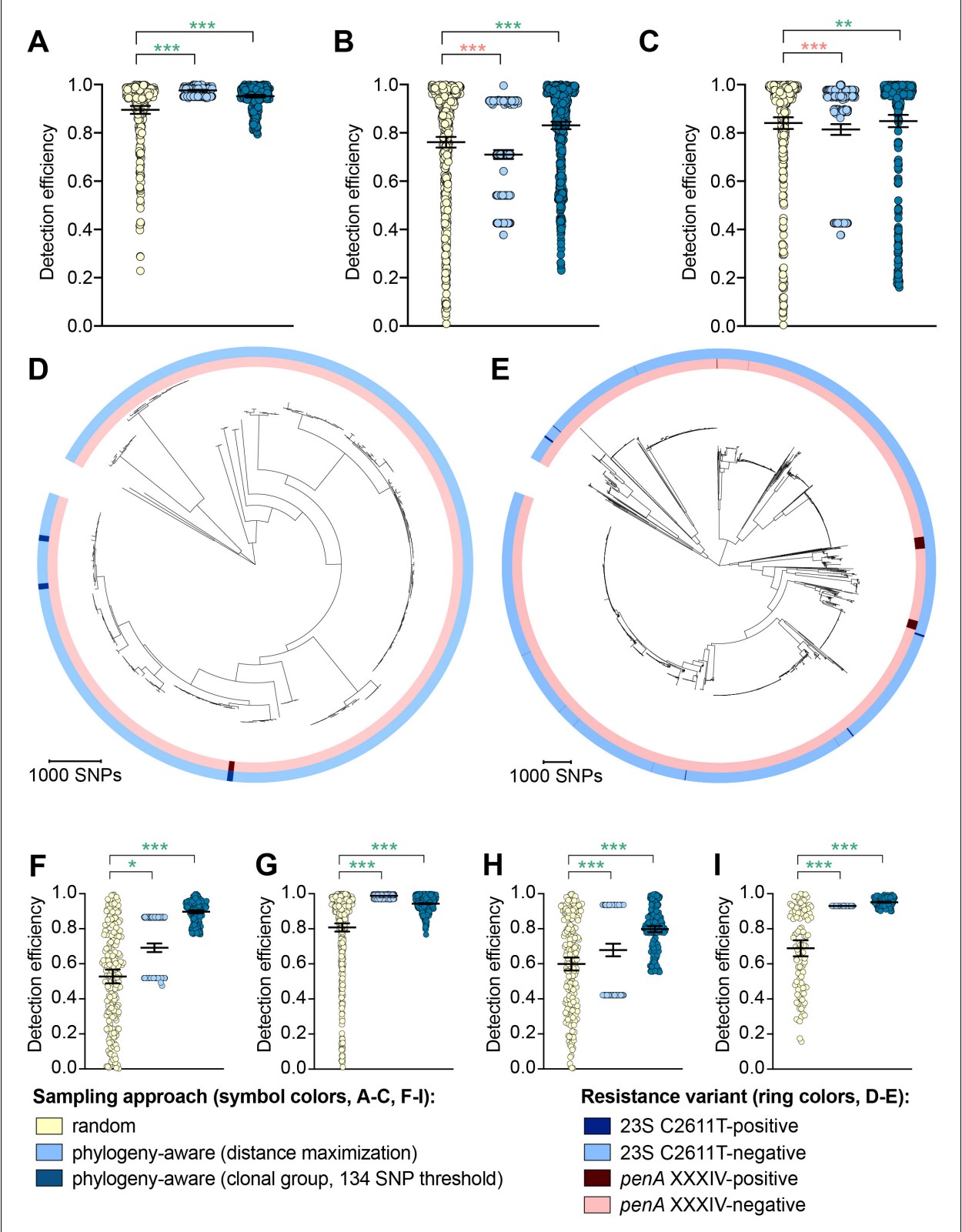

**Figure 2.** The impact of phylogeny-aware sampling on the detection efficiency of genetic resistance and diagnostic escape variants. Scatter dot plots showing the detection efficiency (with lines indicating the mean and 95% confidence intervals from 100 simulations) for resistance variants RplD G70D (**A**), 23S rRNA C2611T (**B**), and *penA* XXXIV (**C**) in datasets 1–5. Note that sampling simulations were not performed for RplD G70D in datasets 1 and 4 or for *penA* XXXIV in dataset 3 as prevalence of the variants in these datasets was >10%. Maximum-likelihood phylogenies produced from

*Figure 2 continued on next page*

*Figure 2 continued*

pseudogenome alignments (with predicted regions of recombination removed) of isolates from dataset 4 (**D**) and dataset 2 (**E**). Presence or absence of the 23S rRNA C2611T mutation (in at least 2/4 alleles) and the mosaic *penA* XXXIV allele is indicated by colored rings. Scatter dot plots showing the detection efficiency (with lines indicating the mean and 95% confidence intervals from 100 simulations) for diagnostic-associated variants 16S rRNA C1209A (**F**), *N. meningitidis*-like *porA* (**G**), *cppB* deletion (**H**), and DR-9A G168A (**I**) in all datasets in which the variant was present. Dot colors in **A–C**) and **F–I**) indicate the sampling approach, and asterisks indicate a significant difference (p<0.05 by Mann-Whitney U test) in detection efficiency between the phylogeny-aware approach compared to random sampling (*p<0.05, **p<0.01, ***p<0.001; red asterisks indicate significantly lower detection efficiency of the phylogeny-aware approach compared to random sampling, and green asterisks indicate significantly higher detection efficiency of the phylogeny-aware approach compared to random sampling). n.s., not significant at α = 0.05.

The online version of this article includes the following figure supplement(s) for figure 2:

**Figure supplement 1.** Detection efficiency of clonal group sampling across different similarity thresholds.

## Targeted sampling based on genetic markers

Multiple drug resistance is more common in pathogenic bacteria than one would expect from the product of frequencies of resistance to individual drugs (*Chang et al., 2015*; *Lehtinen et al., 2019*). This suggests that novel resistance mechanisms might be more likely to arise and spread in bacterial strains that are already resistant to other drugs, a phenomenon that has been documented in *N. gonorrhoeae* (*Goldstein et al., 2012*). It may therefore be fruitful to look for novel resistance variants for one drug in genetic backgrounds that are resistant to other drugs. It may be similarly effective to sample preferentially isolates with genetic markers that have been associated with a range of resistance mechanisms (*e.g.*, through epistatic interactions with other genetic variants) within and/or across different antibiotics when screening for a novel resistance variant. For example, as ciprofloxacin was the recommended first-line therapy for uncomplicated gonorrhea through 2005 in the United Kingdom (*Whittles et al., 2018*), 2007 in the United States (*Centers for Disease Control and Prevention (CDC), 2007*), and more recent years in other countries (*Hemarajata et al., 2016*; *Unemo and Dillon, 2014*; *Bazzo et al., 2018*), we investigated whether resistance to ESCs is significantly more likely to occur in the background of genotypic ciprofloxacin resistance (i.e., in strains with the GyrA S91F mutation). Similarly, as mutations at positions 120 and/or 121 in PorB, the major outer membrane protein in gonococci, have been associated with resistance to a range of drugs from multiple classes (*Mortimer and Grad, 2019*), we investigated whether resistance to ESCs is significantly more likely to occur in strains with PorB 120 and/or 121 mutations. Isolates with CRO-RS and CFX-R were significantly more likely to have the GyrA S91F mutation and the PorB G120 and/or A121 mutations than the wild-type GyrA S91 or wild-type PorB G120/A121 (p<0.001, Fisher's exact test, *Figure 3A–B*). Further, both GyrA S91F and PorB G120 and/or A121 mutations occurred across a range of ESC resistance locus haplotypes (*Figure 3C–D*). For all datasets with CRO-RS or CFX-R isolates, detection efficiency of both variants was significantly increased by only sampling isolates with the GyrA S91F mutation or the PorB G120 and/or A121 mutations (*Figure 3E–F*, *Supplementary file 2F*). Together, these results suggest that preferential sampling of isolates with certain genetic markers, including markers of resistance to previous first-line antibiotics, may increase the detection efficiency of novel resistance variants.

**Table 3.** Summary of the potential diagnostic escape variants assessed.

| Variant | Diagnostic assay | Documented association with diagnostic failure | Prevalence in dataset | | | | |
| --- | --- | --- | --- | --- | --- | --- | --- |
| | | | 1 | 2 | 3 | 4 | 5 |
| 16S rRNA C1209A (four alleles) | Aptima GC Combo | Yes (*Guglielmino et al., 2019*) | 0.11% | 0.09% | 0% | 0% | 0% |
| *N. meningitidis*-like *porA* | In-house (*Whiley et al., 2004*; *Whiley et al., 2005*) | Yes (*Whiley et al., 2011*; *Golparian et al., 2012*) | 0.11% | 0.05% | 0% | 0% | 0% |
| *cppB* deletion | In-house (*Diemert et al., 2002*; *Van Dyck et al., 2001*) | Yes (*Bruisten et al., 2004*) | 1.12% | 0.05% | 0.47% | 0% | 7.29% |
| DR-9A G168A | Roche COBAS 4800 CT/NG | No | 0% | 0.09% | 0% | 0% | 0% |

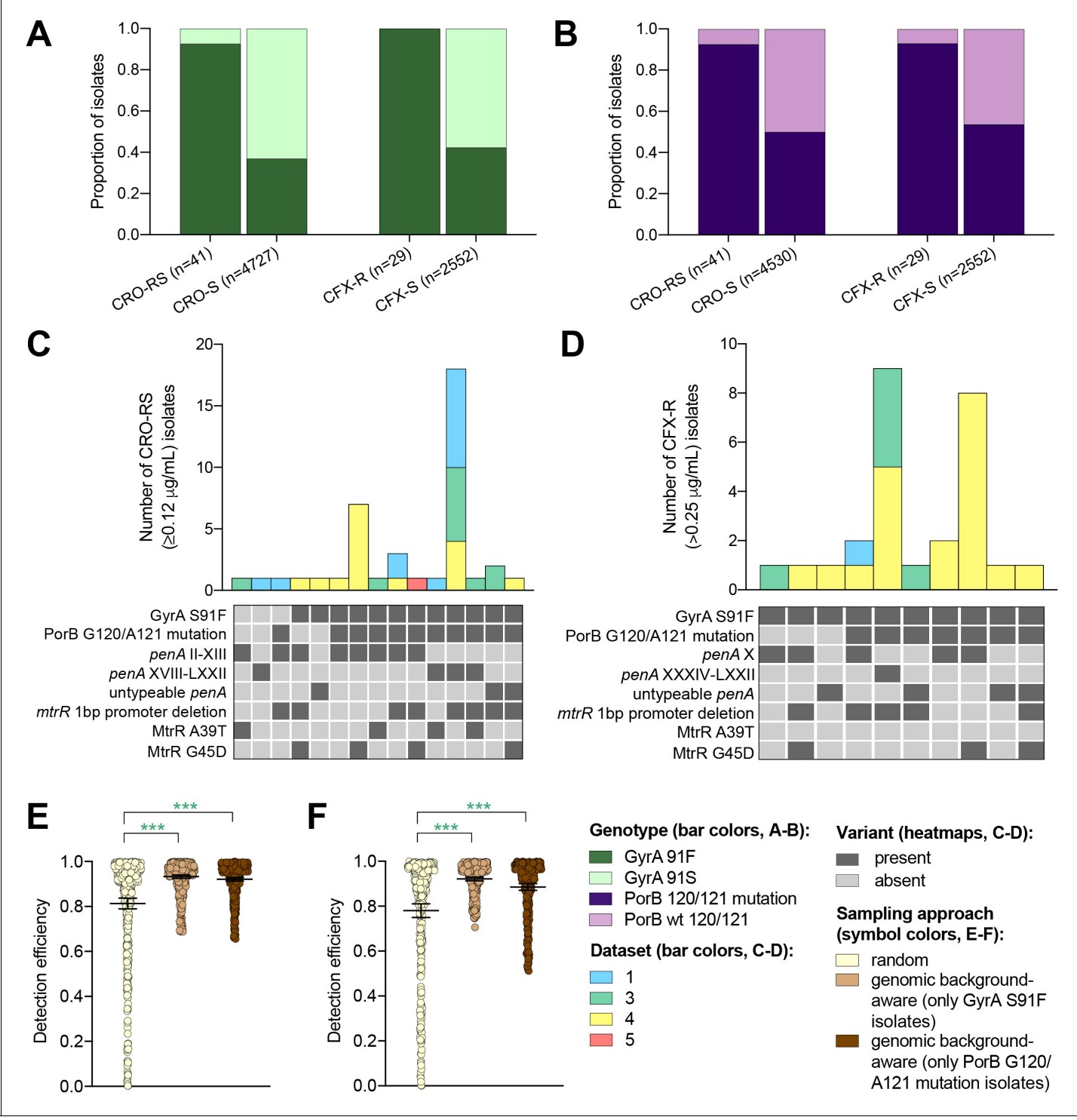

**Figure 3.** The impact of genomic background-aware sampling on the detection efficiency of phenotypic resistance variants. Bar charts showing the proportions of ceftriaxone reduced susceptibility (CRO-RS) isolates, ceftriaxone susceptible (CRO-S) isolates, cefixime resistant (CFX-R) isolates, and cefixime susceptible (CFX-S) isolates with GyrA S91F and GyrA S91 wild-type alleles (**A**) and with PorB G120 and/or A121 mutations and PorB G120 and A121 wild-type alleles (**B**) across datasets 1–5. Bar charts showing the number of (**C**) CRO-RS and (**D**) CFX-R isolates with each haplotype, along with heatmaps showing the presence or absence of the GyrA S19F mutation, the PorB G120 and/or A121 mutations, and other alleles at loci previously associated with extended spectrum cephalosporin resistance. Bar colors in (**C**) and (**D**) indicate the dataset from which the isolates were derived. Scatter dot plots showing the detection efficiency (with lines indicating the mean and 95% confidence intervals from 100 simulations) for CRO-RS (**E**) and CFX-R (**F**) in all datasets in which the variant was present. Dot colors in **E–F**) indicate the sampling approach, and asterisks indicate a significant difference

*Figure 3 continued on next page*

*Figure 3 continued*

(p<0.05 by Mann-Whitney U test) in detection efficiency between the phylogeny-aware approach compared to random sampling (*p<0.05, **p<0.01, ***p<0.001; green asterisks indicate significantly higher detection efficiency of the genomic background-aware approach compared to random sampling).

## Discussion

With sequencing becoming more integral to routine pathogen surveillance and diagnostics, it is important to ensure that models mapping genotypic information to expected pathogen phenotype and/or clinical outcome are comprehensive and current (*Donà et al., 2017*). In the case of genotype-based diagnostics, sustained phenotypic surveillance is crucial for identifying resistance variants that have recently emerged and/or increased in prevalence from previously undetected levels. While effective incorporation of patient metadata into surveillance strategies may be challenging, availability and incorporation of information on pathogen characteristics (*e.g.*, pathogen genomic data) into surveillance programs may ultimately decrease the cost of surveillance to maintain the sensitivity of these diagnostic tools.

Collection of patient metadata, including demographic and geographic information, is crucial to understanding the epidemiology of drug resistance. However, it may be difficult to obtain data on the relevant patient features, and the predictive power of such features may rapidly decay because of patient mobility and interactions (*Goldstein et al., 2017*). While availability of patient metadata varied across the datasets assessed, our results suggest that while incorporation of patient metadata into sampling strategies may increase detection efficiency for some novel resistance variants, it may be difficult to generalize for all potential novel resistance variants. It is possible that targeted sampling based on patient characteristics may be more reliable in the context of pathogens, antibiotic, and/or patient characteristics not assessed here.

Incorporation of WGS into routine pathogen surveillance by public health agencies (*European Centre for Disease Prevention and Control, 2019*; *Brown et al., 2019*) may facilitate use of genomic information in phenotypic sampling strategies, particularly with emerging metagenomic approaches that do not require bacterial culture (*Břinda et al., 2020*). Our results show that phylogeny-aware sampling, particularly the clonal group approach, which reduces the amount of repeated sampling of closely related isolates, significantly improved detection efficiency over random sampling for multiple resistance and diagnostic-associated variants. Further, identification of and preferential sampling of isolates with genetic markers that are consistently predictive of resistance across a range of mechanisms, including those associated with resistance to other drugs, may supplement phylogeny-aware sampling to further optimize detection efficiency of novel variants. However, the utility of sampling based on genetic markers of other resistance mechanisms will likely vary substantially across different drugs and be influenced by future treatment guidelines.

While the clonal group sampling approach increased detection efficiency for the resistance and diagnostic escape variants assessed here, it may be difficult to determine the most effective and reliable metric or threshold for clonal grouping, especially as this is likely to vary across different clinical populations, antibiotics, and bacterial species. Detection efficiency was generally consistent across the two SNP thresholds and fastBAPS groupings based on WGS. However, performance of the clonal group approach using MLSTs was inconsistent and, in some instances, worse than random sampling, likely due to the shortcomings of MLST compared to WGS-based approaches in distinguishing between AMR variant-positive clades and more distantly-related variant-negative clades in species such as *N. gonorrhoeae* (*Harris et al., 2018*). This suggests that this approach is sensitive to similarity thresholds and that a low SNP threshold based on WGS assemblies may be the most appropriate approach, particularly in a population where there is expected to be substantial clonality among isolates and thus, even with a low threshold, detection efficiency will be improved by the clonal group approach. More broadly, surveillance incorporating WGS rather than MLST loci alone may further promote NAAT sustainability by enabling screening for variants with previously undetected mutations in target loci, such as the *N. gonorrhoeae* DR-9A G168A variants, that may be associated with diagnostic escape. While sequencing errors may occasionally impair clonal grouping of closely related isolates, thus weakening the benefit of the clonal group approach relative to

random sampling, we expect the impact of such errors to be marginal in the absence of large sequence quality issues. Such large quality issues should be apparent at the first stages of an analysis.

We have assessed these targeted sampling approaches in detection of multiple resistance variants across a range of populations, but these represent only a fraction of resistance mechanisms in a single species. These findings may extend to other antibiotics and bacterial species. For example, given the high degree of clonality among *M. tuberculosis* isolates and the significant variation in prevalence of drug resistance and resistance-conferring genotypes across clonal groups (*Merker et al., 2015*; *Casali et al., 2014*), the clonal group sampling approach may similarly improve detection efficiency of novel resistance variants in *M. tuberculosis*. For species in which drug resistance is primarily acquired through gene acquisition, it is unclear if phylogeny-aware sampling based on the core genome will improve detection efficiency of novel variants, though in gonococcus, there is evidence of a relationship between the core genome and the plasmid-borne resistance genes *bla*TEM and *tetM* (*Sánchez-Busó et al., 2019*), and it is further possible that, combined with core genome-based phylogeny-informed sampling, screening for homologs of known resistance genes from other species may expedite identification of any novel resistance genes acquired by the species of interest. However, in addition to providing a more practically applicable (i.e., less computationally intensive) alternative to phylogeny-informed sampling, sampling informed by k-mer distances (*Ondov et al., 2016*; *Lees et al., 2019*) may also be more generalizable to a broader range of novel resistance mechanisms. Further, the requirement of confirmatory phenotyping to identify novel resistance may not extend to pathogens that are expected to be associated with reliably-identifiable treatment failures, as for these pathogens, identification of treatment failure likely represents the most efficient method of novel resistance variant detection (*Berenger et al., 2019*). However, for other pathogens, such as *N. gonorrhoeae* (*Eyre et al., 2018*), treatment failures may go undetected for reasons including partial abatement of symptoms or long treatment regimens. Ultimately, as genotype-based diagnostics for antibiotic resistance become available for more species, it will be important to assess the efficiencies of these approaches across pathogens with different clinical, epidemiological, and evolutionary paradigms.

Since we lack the datasets to assess targeted sampling of variants from the time they first emerged in a population, any associations we observed between the variants and patient or pathogen features do not necessarily reflect those around the time of emergence. Thus, more longitudinal epidemiological and genomic studies, particularly after the implementation of genotype-based diagnostics, are necessary to better characterize patterns of novel resistance emergence and inform targeted surveillance approaches.

The phylogeny-aware sampling approaches presented here are based on the assumption that genomic data will be available for the pool of potential isolates from incident cases that may undergo confirmatory phenotyping. However, using information on isolate features to increase surveillance efficiency may be feasible even in the absence of mass prospective sequencing. For example, under the general assumption that novel resistance variants are more likely to appear in underrepresented lineages, phylogeny-aware surveillance could be paired with a diagnostic approach such as genomic neighbor typing (*Břinda et al., 2020*), where any isolates with either susceptible or low confidence calls that appear to be divergent from the genomes in the reference database would be prioritized for confirmatory phenotyping. Similarly, a diagnostic that predicts AMR phenotypes through a combination of transcriptomic and genomic typing (*Bhattacharyya et al., 2019*) may facilitate targeted surveillance by identifying isolates with ambiguous predictions (*e.g.*, isolates with transcriptional signatures of resistance that lack known genomic markers of resistance) that could be prioritized for confirmatory phenotyping.

While the focus of this study was to introduce and evaluate approaches to increase the efficiency of surveillance programs for maintaining marker-based AMR diagnostics, these approaches may be broadly applicable to surveillance programs aimed at tracking AMR in general and/or other phenotypes of interest that may be time- and/or resource-intensive to directly measure. For example, programs such as the National Antimicrobial Resistance Monitoring System for Enteric Bacteria (https://www.cdc.gov/narms/) could adopt these targeted sampling approaches to prioritize isolates for phenotypic testing.

Advances in diagnostics, extensive sequencing of clinical isolates, and large collections of clinical and pathogen data together provide new opportunities for integrating data streams and optimizing

surveillance efforts. As marker-based point-of-care AMR diagnostics are developed and implemented, optimization of surveillance systems will require assessments like those modeled here of species-, drug-, and population-specific factors that may affect the emergence and distribution of diagnostic escape resistance variants, as well as how the diagnostic itself may complement surveillance efforts.

## Materials and methods

### Dataset preparation and phylogenetic reconstruction

See *Table 1* for details of the *N. gonorrhoeae* datasets and *Tables 2* and *3* for the variants assessed. Raw sequencing data were downloaded from the NCBI Sequence Read Archive. Genomes were assembled using SPAdes v3.13 (*Bankevich et al., 2012*) with default parameters and the careful option to minimize the number of mismatches. Assembly quality was assessed using QUAST v4.3 (*Gurevich et al., 2013*), and contigs < 500 bp in length and/or with <10 x average coverage were removed. Isolate reference-based pseudogenomes were constructed by mapping raw reads to the NCCP11945 reference genome (RefSeq accession number NC_011035.1) using BWA-MEM v7.12 (*Li, 2013*), the Picard toolkit v2.8 (*Picard development team, 2016*) to identify duplicate reads, and Pilon v1.22 (*Walker et al., 2014*) to determine the base call for each site, with a minimum depth of 10 and a minimum base quality of 20.

Loci in *Tables 2* and *3* were extracted from the genome assemblies using blastn (*Altschul et al., 1990*) followed by MUSCLE alignment using default parameters (*Edgar, 2004*) to assess the presence or absence of the resistance variants. Presence or absence of mutations in the multi-copy 16S and 23S rRNA genes and the repetitive DR-9A and DR-9B regions (*Dailey et al., 2013*) was assessed using BWA-MEM, the Picard toolkit, and Pilon, as above, to map raw reads to a single 16S rRNA allele, a single 23S rRNA allele, a single DR-9A region, and a single DR-9B region from the NCCP11945 reference isolate and determine the mapping quality-weighted percentage of each nucleotide at the site of interest. See *Table 4* for information on the reference sequences used for variant calling. Isolate metadata and resistance variant profiles are given in *Supplementary file 1*. Sampling bias across demographic and geographic groups was assessed by comparing (by chi-squared test) reported gonorrhea incidence across the groups in the population from which the dataset samples were collected to the prevalence of the groups in each dataset. Association between genetic variants and demographic or geographic groups and between phenotypic resistance variants and genetic markers in each of datasets was assessed by Fisher's exact test, due to the low prevalence of the variants.

Gubbins v2.3.4 (*Croucher et al., 2015*) was used with default parameters to identify and mask recombinant regions from the pseudogenomes and build maximum likelihood phylogenies from the

**Table 4.** Reference information for the genetic variants assessed.

| Variant | Reference accession | Coordinates of genetic locus in reference entry | Position of mutation in reference locus |
| --- | --- | --- | --- |
| RplD G70D | NC_011035.1 | 2033052–2033672 | amino acid 70 |
| 23S rRNA C2611T | NC_011035.1 | 1263408–1266305 | nucleotide 2603 |
| *penA* XXXIV | NZ_LT906440.1 | 1588456–1590201 | N/A (assessed presence/absence of this allele) |
| 16S rRNA C1209A | NC_011035.1 | 1266903–1268450 | nucleotide 1192 |
| *N. meningitidis*-like *porA* | NC_011035.1 | 735796–737125 | N/A (assessed nucleotide similarity across the full locus with a threshold of ≤ 90%)[*] |
| *cppB* deletion | LT592149.1 | 2912–3553 | N/A (assessed presence/absence of full locus) |
| DR-9A G168A | NC_011035.1 | 530088–530277 | nucleotide 168 |

[*]Isolates with a *porA* pseudogene with ≤90% similarity to the NC_011035.1 *porA* pseudogene were called positive for *N. meningitidis*-like *porA*. Note that all such isolates were confirmed to have a *porA* pseudogene that was ≥92% similar to the *N. meningitidis porA* (GenBank Accession: GQ173789.1), while all other isolates had ≤89% similarity to the *N. meningitidis porA*.

non-recombinant pseudogenome alignments for each dataset through RAxML v8.2.12 (*Sommer et al., 2017*). Pairwise phylogenetic distances were calculated after removal of predicted recombinant regions using the ape package in R. Phylogenetic distributions of genetic resistance variants were assessed by estimating the phylogenetic D statistic (*Fritz and Purvis, 2010*) using the caper package in R. Bayesian analysis of population structure was performed on the pseudogenome alignments for each dataset using fastBAPS (*Tonkin-Hill et al., 2019*). Multilocus sequence types (MLSTs) were assigned using the PubMLST database (https://pubmlst.org/neisseria/).

## Sampling approaches

For each sampling approach/dataset/variant combination, 100 simulations were carried out with isolate sampling continuing until variant detection. We defined 'detection efficiency' as one minus the fraction of isolates sampled prior to variant detection (excluding any samples for which the presence or absence of the variant could not be determined). As detection efficiencies were not normally distributed, differential performance between random sampling and targeted sampling was assessed by Mann Whitney U tests of differences in mean ranks of detection efficiencies. Because the purpose of this study was to compare the rare variant detection efficiency between random sampling and targeted sampling approaches, we did not evaluate RplD G70D in datasets 1 and 4 or for the *penA* XXXIV allele in dataset 3, as the prevalence of these variants in these datasets was >10%.

In demography-aware sampling (datasets 1 and 2), the first isolate was selected at random, and each successive isolate was randomly selected from alternating demographic groups (men vs. women and men who have sex with men [MSM] vs. men who have sex with women [MSW] or women who have sex with men [WSM]). For anatomical site (niche)-aware sampling (datasets 1 and 2), the first isolate was selected at random, and each successive isolate was randomly selected from alternating anatomical sites of isolate collection (i.e., cervix, urethra, rectum, and pharynx). For geography-aware sampling (datasets 3 and 4), the first isolate was selected at random, and each successive isolate was randomly selected from alternating geographic regions (countries or prefectures). For geography- and distance-aware sampling (datasets 3 and 4), the first isolate was selected at random, and each successive isolate was selected randomly from the region (country or prefecture) with the largest product of geographic distances from previously sampled regions, only re-sampling from a given region after all regions had been sampled in that round. For travel history- and sex work-aware sampling (dataset 2), isolates were selected at random either limiting the pool to isolates from patients who had recently engaged in overseas sex or sex work, respectively (*Williamson et al., 2019*).

For phylogeny-aware sampling (datasets 1-5), the first isolate was selected at random, and each successive isolate was either selected to maximize the product of phylogenetic distances from each of the previously sampled isolates ("distance maximization") or selected randomly with the exception of ensuring even sampling across phylogenetic groups ("clonal group"; i.e., isolates $\leq N$ SNPs from a previously sampled isolate that were excluded from future sampling until all "clonal groups" had been sampled). SNP cutoffs tested for the clonal group approach included 1) 134 SNPs, the lower 95% confidence interval of the mean SNP distance across datasets 1-5 between each isolate with phenotypic cefixime resistance (CFX-R), azithromycin resistance (AZM-R), and/or ceftriaxone reduced susceptibility (CRO-RS, >0.25 μg/mL, >1 μg/mL, and ≥0.12 μg/mL, respectively) and the closest susceptible isolate, and 2) 422 SNPs, the lower 95% confidence interval of the mean SNP distance across datasets 1-5 between each isolate with the RplD G70D mutation, the 23S rRNA C2611T mutation, and/or the *penA* XXXIV allele and the closest isolate without the resistance variant. The clonal group sampling approach was further tested by alternating sampling across fastBAPS and MLST groups.

For genomic background-aware sampling, isolates were selected at random either limiting the pool to isolates with genotypic ciprofloxacin resistance (i.e., the GyrA S91F mutation) or to isolates with a mutation at PorB G120 and/or PorB A121, which have been associated with a range of resistance pathways in multiple classes of antibiotics (*Mortimer and Grad, 2019*). Genomic background-aware sampling was assessed in detection of CRO-RS (datasets 1 and 3–5; dataset 2 had no CRO-RS isolates) and CFX-R (datasets 1 and 3–4; datasets 2 and 5 had no CFX-RS isolates).

## Additional information

### Competing interests

Marc Lipsitch: Reviewing editor, *eLife*. The other authors declare that no competing interests exist.

### Funding

| Funder | Grant reference number | Author |
|---|---|---|
| National Institute of Allergy and Infectious Diseases (NIAID) | R01AI132606 | Yonatan H Grad |

The funders had no role in study design, data collection and interpretation, or the decision to submit the work for publication.

### Author contributions

Allison L Hicks, Yonatan H Grad, Conceptualization, Resources, Supervision, Funding acquisition, Methodology, Writing - original draft, Writing - review and editing; Stephen M Kissler, Methodology, Writing - original draft, Writing - review and editing; Tatum D Mortimer, Data curation, Formal analysis, Methodology, Writing - review and editing; Kevin C Ma, Data curation, Formal analysis, Writing - review and editing; George Taiaroa, Data curation, Formal analysis, Writing - original draft, Writing - review and editing; Melinda Ashcroft, Deborah A Williamson, Data curation, Writing - review and editing; Marc Lipsitch, Conceptualization, Data curation, Methodology, Writing - original draft, Writing - review and editing

### Author ORCIDs

Allison L Hicks https://orcid.org/0000-0003-1372-1301
Stephen M Kissler https://orcid.org/0000-0003-3062-7800
Tatum D Mortimer https://orcid.org/0000-0001-6255-690X
Deborah A Williamson https://orcid.org/0000-0001-7363-6665
Marc Lipsitch https://orcid.org/0000-0003-1504-9213
Yonatan H Grad https://orcid.org/0000-0001-5646-1314

### Decision letter and Author response

Decision letter https://doi.org/10.7554/eLife.56367.sa1
Author response https://doi.org/10.7554/eLife.56367.sa2

## Additional files

### Supplementary files

• Supplementary file 1. Metadata and resistance variant profiles for isolates assessed in this study. M, men; W, women; MSM, men who have sex with men; MSW, men who have sex with women; WSM, women who have sex with men; CFX-R, cefixime resistance; CRO-RS, ceftriaxone reduced susceptibility; AZM-R, azithromycin resistance.

• Supplementary file 2. Dataset bias and targeted sampling results. (A) Demographic and geographic sampling biases in datasets 1–3. (B) Detection efficiency of random, demography-, niche-, and geography-aware sampling approaches for resistance variants. (C) Detection efficiency of random sampling, as well as preferential sampling of patients that had recently engaged in overseas sex or in sex work, for resistance variants in dataset 2. (D) Detection efficiency of random and phylogeny-aware sampling approaches for resistance variants. (E) Detection efficiency of random and phylogeny-aware sampling approaches for variants associated with diagnostic escape. (F) Detection efficiency of random and genomic background-aware sampling approaches for resistance variants.

• Transparent reporting form

## Data availability

The source data for all figures and tables are included in available in Supplementary file 1 and/or the NCBI Sequence Read Archive (BioProject numbers indicated in Table 1 and individual sample accession numbers indicated in Supplementary file 1).

The following previously published datasets were used:

| Author(s) | Year | Dataset title | Dataset URL | Database and Identifier |
|---|---|---|---|---|
| Williamson D, Chow EPF, Gorrie C, Seemann T, Ingle DJ, Higgins N | 2019 | Bridging of Neisseria Gonorrhoeae Across Diverse Sexual Networks in the HIV Pre-Exposure Prophylaxis (PrEP) Era: A Clinical and Molecular Epidemiological Study | https://www.ncbi.nlm.nih.gov/sra/?term=SRP185594 | NCBI Sequence Read Archive, SRP185594 |
| Harris SR, Cole MJ, Spiteri G, Sanchez-Buso L, Golparian D, Jacobsson S | 2018 | Public health surveillance of multidrug-resistant clones of Neisseria gonorrhoeae in Europe: a genomic survey | https://www.ncbi.nlm.nih.gov/sra/?term=ERP010312 | NCBI Sequence Read Archive, ERP010 312 |
| Yahara K, Nakayama SI, Shimuta K, Lee KI, Morita M, Kawahata T | 2018 | Genomic surveillance of Neisseria gonorrhoeae to investigate the distribution and evolution of antimicrobial-resistance determinants and lineages. | https://www.ncbi.nlm.nih.gov/sra/?term=DRP004052 | NCBI Sequence Read Archive, DRP0040 52 |
| Lee RS, Seemann T, Heffernan H, Kwong JC, Goncalves da Silva A, Carter GP | 2018 | Genomic epidemiology and antimicrobial resistance of Neisseria gonorrhoeae in New Zealand | https://www.ncbi.nlm.nih.gov/sra/?term=SRP111927 | NCBI Sequence Read Archive, SRP111927 |
| Mortimer TD | 2020 | Using genomics to understand transmission networks of Neisseria gonorrhoeae in New York City | https://www.ncbi.nlm.nih.gov/sra/?term=ERP011192 | NCBI Sequence Read Archive, ERP011192 |

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
