## [Decision Letter]

**Acceptance summary:**

Identification of antibiotic resistance is shifting from culture based to genetic identification of drug resistant strains. However, how best to use these data is unclear – can we move beyond searching for common and known variants to identify new mutations? This manuscript addresses the critical question of how to rapidly identify antibiotic resistance from sequence data. It compares how inclusion of different clinical and genetic data affects these predictions. It finds a limited role for inclusion of clinical data but suggests that inclusion of data on phylogeny and background may be informative.

**Decision letter after peer review:**

Thank you for submitting your article "Targeted surveillance strategies for efficient detection of novel antibiotic resistance variants" for consideration by *eLife*. Your article has been reviewed by two peer reviewers, and the evaluation has been overseen by a Reviewing Editor and Eduardo Franco as the Senior Editor. The following individuals involved in review of your submission have agreed to reveal their identity: Michael Feldgarden (Reviewer #1); Nicholas Medland (Reviewer #2).

The reviewers have discussed the reviews with one another and the Reviewing Editor has drafted this decision to help you prepare a revised submission.

Summary:

This paper addresses a critical shortcoming of the transition from conventional culture-based N. gonorrhoea AMR testing to molecular based systems, which otherwise have the potential to transform both diagnostics and surveillance globally. Determining the extent of the research flatform required support molecular AMR testing and how it is applied to real world sampling is critical. In particular, knowing that resistance develops according to known mechanisms and that population and anatomical distribution is not random, are sampling frames able to be manipulated to increase the probability that new resistant variants will be detected sooner? Using five historical datasets with established genotype and culture-based resistance results but with varying degrees of clinical and epidemiological data (from highly detailed to unavailable), the authors simulated whether different sampling based on demographic, anatomic and/or geographic sampling frames would detect markers of clinically important azithromycin and ceftriaxone resistance known to occur in low prevalence in those datasets. They found that although some targeted sampling, in particular of returned travellers, helps to identify resistant mutants, on the whole it did not do so reliably.

So as to better target sampling at strains that are genetically different from those previously sampled the authors propose two phylogenetic strategies: sampling maximally unrelated isolates and sampling from each phylogenetically related clone. These approaches led to some improvement in efficiency in detecting resistant mutants, particularly after post-hoc adjustment of sensitivity thresholds (for the latter) and in some data sets with particular characteristics (for the former).

A third type of strategy was to sample strains what had already identified markers of resistant to other antibiotics, in particularly to ciprofloxacin whose resistance genotype is extremely well studied and well characterised.

The first approach uses metadata, which is considered essential in surveillance but is nonetheless not always available, and which the later approaches have the advantage of not requiring. However, the second and third approaches clearly require some preliminary molecular testing of specimens to determine which to sample for whole genome sequencing.

Essential revisions:

1) How would a low rate of either sequencing or phenotyping error affect the different approaches? Presumably, these effects would not qualitatively affect the outcomes, but would such error rates, which are often observed, falsely improve or weaken one approach?.

2) The resistance markers used are mostly point mutations or recombinant alleles. What are the expectations for the performance of these sampling approaches for acquired resistance genes (e.g., plasmid borne AMR genes)?

3) There are other large datasets with genomic and phenotypic information (e.g., NARMS) that could be used to test this-and find this approach useful. While I do not expect (nor want) the authors to examine these datasets, recognizing that this technique would be useful for large surveillance systems should be mentioned. It would be a shame for surveillance systems to be unaware of these findings.

---

## [Author Response]

Essential revisions:1) How would a low rate of either sequencing or phenotyping error affect the different approaches? Presumably, these effects would not qualitatively affect the outcomes, but would such error rates, which are often observed, falsely improve or weaken one approach?.

Thank you for bringing this up and giving us the opportunity to clarify and address this.

While false-positive phenotyping errors (i.e., a susceptible isolate that was called resistant) would presumably be further investigated and ultimately corrected, we agree that false-negative phenotyping errors (i.e., a resistant isolate that was called susceptible) could absolutely delay detection of a novel AMR variant. However, we don’t believe there is reason to think that this should disproportionately affect detection efficiency of any of the targeted sampling approaches compared to random sampling or compared to each other.

Sequencing errors could affect the efficiency of the phylogeny-aware sampling approaches. The differential prevalence of resistance variants among datasets might raise the concern that the improved detection efficiency by phylogeny-aware sampling arose from study-dependent sequencing artifacts. However, we evaluated each dataset separately, so variation in sequencing methods across datasets should not have confounded our analyses. When applied in a surveillance setting, we do agree that sequencing errors could in theory result in closely related isolates appearing to be more distantly related, thus reducing the increased efficiency of phylogeny-aware sampling compared to random sampling. However, we don’t expect that the opposite would occur (i.e., it is unlikely that sequencing errors would result in distantly related isolates appearing to be very closely related), so sequencing errors are unlikely to make the clonal group approach worse than random sampling. Further, we expect that the influence of sequencing errors should be largely ameliorated by inspecting sequencing quality prior to analysis. We have now added some discussion of sequencing errors (Discussion, fourth paragraph).

2) The resistance markers used are mostly point mutations or recombinant alleles. What are the expectations for the performance of these sampling approaches for acquired resistance genes (e.g., plasmid borne AMR genes)?

Thank you for bringing up this point. We agree that it is not entirely clear how phylogeny-aware sampling (particularly when based on reference-based mapping, as we have done here) will perform in detecting novel variants associated with gene acquisition. While the gonococcal resistance variants associated with plasmids are too prevalent to be useful for our evaluations here, there is some evidence for a relationship between the core genome and the presence/absence of these plasmids, suggesting that core genome-based phylogeny-aware sampling may still be useful for these kinds of resistance variants. We also suspect that with the genomic data required for phylogeny-informed sampling, it may be possible to rapidly identify some novel resistance by screening for homologs of known resistance genes from other species. Further, k-mer based methods that allow for clustering based on both core and accessory genome similarity may be more useful in the context of novel resistance associated with gene acquisition. We have now expanded our discussion of this (Discussion, fifth paragraph).

3) There are other large datasets with genomic and phenotypic information (e.g., NARMS) that could be used to test this-and find this approach useful. While I do not expect (nor want) the authors to examine these datasets, recognizing that this technique would be useful for large surveillance systems should be mentioned. It would be a shame for surveillance systems to be unaware of these findings.

Thank you for this suggestion. We agree that these targeted sampling approaches may be more broadly applicable outside of the context of maintaining diagnostics. We have now highlighted this in the Discussion (eighth paragraph).